# Borate–Guanosine Hydrogels and Their Hypothetical Participation in the Prebiological Selection of Ribonucleoside Anomers: A Computational (DFT) Study

**DOI:** 10.3390/ijms262412103

**Published:** 2025-12-16

**Authors:** Ana Franco, Adelino M. Galvão, José A. L. da Silva

**Affiliations:** Centro de Química Estrutural, Institute of Molecular Sciences, Departamento de Engenharia Química, Instituto Superior Técnico, Universidade de Lisboa, 1049-001 Lisboa, Portugal; anacfbanha@tecnico.ulisboa.pt

**Keywords:** borate, guanosine, G4-quartets, hydrogel, anomeric selectivity, RNA world hypothesis, origin of life, density functional calculations

## Abstract

The prebiological anomeric selectivity of ribonucleosides is a key phenomenon in the understanding of the RNA world hypothesis and the origin of life. While each ribonucleoside can have two anomers (α or β), ribonucleosides naturally occur in the β form, while α anomers are extremely rare. Guanosine, a canonical ribonucleoside, binds to borate and self-assembles into G4-quartets, enabling the formation of borate–guanosine hydrogels. These macrostructures, exhibiting elevated thermal robustness and self-healing properties, have been suggested as plausible frameworks for the syntheses of prebiological molecules. Moreover, their external layers could have prevented degradation of compounds by aggressive primitive radiation and reduced molecular dispersion. Herein it is proposed that anomeric selectivity may have occurred due to the different 3D organization and stereochemical environment formed by each borate–guanosine anomer and subsequent formation of borate–guanosine hydrogels. DFT was applied to the optimization of α and β anomeric structures in four steps, from borate–guanosine diesters to G4 structures. The results obtained suggest that β-syn-guanosine (the most stable structure) is the only anomer that forms a planar G4-quartet with borate, capable of self-assembling into a hydrogel. Given the properties of borate–guanosine hydrogels, this could explain why β-guanosine is currently the sole anomer present in living organisms.

## 1. Introduction

Canonical ribonucleoside guanosine is a common building block for supramolecular hydrogels due to its distinctive properties [1]. Its guanine core enables the formation of multiple hydrogen bonds, sometimes leading to self-assembled structures, like G4-quartets [2]. This structure exhibits a central hollow, where four carbonyl oxygen atoms can coordinate with an alkali cation, forming column-like aggregates of G4-quartets through π-π stacking, immobilizing large amounts of water, and hence producing hydrogels [3].

Borate–guanosine hydrogels are particularly interesting since they are more stable [3] than guanosine and 5′-guanosine monophosphate analogues [1,2,4,5,6,7,8,9,10,11,12,13,14,15,16]. In these, borate binds to guanosine at positions 2′ and 3′, forming covalent diesters [3,17] (Figure 1). During self-assembly and stacking, the column-like G4-quartet stacks bind laterally to each other, through the borate diesters, to form entangled fibers, making these hydrogels suitable for several biological applications [1,2,18,19,20,21,22,23,24,25,26,27,28,29]. Moreover, experimental data show that K^+^ greatly improves the self-assembly of borate–guanosine hydrogels, enhancing their stability and robustness when compared with other alkali cations [3].

Given their properties, hydrogels have been proposed as possible frameworks for prebiotic reactions [30,31,32,33], including borate–guanosine ones [18]. Borate is considered an important prebiological reagent [17], since it mediates the abiotic synthesis of ribose (the sugar frame in RNA), ribonucleosides, and ribonucleotides [17]; forms esters with ribose in aqueous and dry media, stabilizing it within a reasonable range of temperature and hindering its degradation [17,34,35]; and induces the occurrence of ribofuranose, the five-membered ring conformation present in biological derivatives (but only present in small amounts in aqueous solution in the absence of borate) [17]. Additionally, since borate binds to ribonucleosides through positions 2′ and 3′, it protects these hydroxyl groups, preventing side phosphorylation reactions, an important condition for the prebiological synthesis of 5′-ribonucleotides and RNA [17]. Previous studies show that borate–guanosine hydrogels exhibit thermal robustness over a large temperature range and tend to self-assemble even when guanosine is phosphorylated (forming ribonucleotides) [6]. Additionally, their external layers could have been primitive membranes of preliminary protocells, which could have prevented the degradation of compounds by aggressive radiation (as happened in primitive Earth) [30,31], confined prebiotic molecules (like other ribonucleosides/ribonucleotides [2,17]) and reduced molecular dispersion (facilitating other prebiological reactions) [17].

Interestingly, the β-anomer of guanosine is predominant in biological systems, whereas the α anomer is extremely rare. While β-guanosine tends to self-assemble into hydrogels (Figure 1), it is uncertain if the same could happen to α-guanosine [17,36]. Given all the properties of borate–guanosine hydrogels, it is possible that these macrostructures could have contributed to the separation of the two anomers, which could have led to the subsequent selective synthesis of β-guanosine in living beings [17]. This separation may have occurred due to the different 3D organization and stereochemical environment of each anomer.

Notably, computational approaches to prebiological chemistry and the origin of life have been previously used with success [37], e.g., several studies focusing on the prebiotic synthesis of reagents [38,39,40]; on the interaction between amino acids, nucleobases or ribonucleosides, and mineral surfaces [41,42,43]; or on the stability of borate–aldopentose complexes [44] have been published. This study explores the structure of borate–guanosine G4-quartets and their possible role in the separation of guanosine anomers, focusing on the analysis and interpretation of computational geometry optimizations of the DFT type.

## 2. Results and Discussion

Each G4-quartet was built step by step, starting with the geometry optimization of the simplest structure possible: a borate–guanosine diester (Figure 2). The most stable diester (of each combination) was used to build and optimize its respective G2, by adding an equal diester to the originally optimized structure. The same procedure was used to obtain its respective G3 and G4. This method was applied to every diester herein presented.

In step **1**, it was considered that borate can form two types of diesters with guanosine (1 and 2) [2,3] (Figure 3A) and that, in addition to anomers α and β, guanosine can have two conformations, anti or syn (Figure 3B).

The α anomer occurs when the nucleobase on the anomeric carbon is *trans* to the -CH_2_OH group, whereas the β anomer ensues that the nucleobase is *cis* to the -CH_2_OH group. Furthermore, the nucleobase can rotate around the β N9 glycosidic bond of guanosine, either projecting away from the ribofuranose ring (anti conformation) or toward the ring (syn conformation, indicated by **S**). Usually the anti conformation is favored [45], but previous studies indicate that the syn conformation is preferred in borate–guanosine hydrogels [46]. Henceforth, the first letter of the name of each structure indicates its conformation: anti (**A**) or syn (**S**).

To simplify, diester geometry optimization was restricted to structures with two anti or two syn guanosines. Diesters of type 1 and 2 (henceforth indicated in subscript) were built with three combinations:(1)α diesters, with two α-guanosines (types α_1_ and α_2_);(2)β diesters, with two β-guanosines (types β_1_ and β_2_);(3)γ diesters, with one α-guanosine and one β-guanosine (types γ_1_ and γ_2_).

Table 1 lists the stability of anti and syn borate–guanosine diesters relative to the most stable structures for each conformation, **Aα_2_** and **Sβ_1_**, respectively (Figure 4A). Calculated energy differences show that **Aγ_1_** and **Sγ_2_** are the most stable diesters of the γ combination. On the other hand, **Aβ_1_** and **Sα_1_** are the most stable structures of the anti β and syn α diesters, respectively. Table 1 shows that two syn α-guanosines form the less stable diesters, while syn β are the most stable diesters (lowest values of ΔE). It is also important to note that while **Aα_2_** is one of the most stable structures, it seems unlikely to form a plane G4-quartet, since it bends over on itself (compare **Aα_2_** diester in Figure 4A with Figure 2). A similar effect is noticeable in the remaining **Aβ_1_**, **Aγ_1_**, and **Sγ_2_** diesters (Appendix A, respectively; for individual geometry-optimized structures of each diester, see Appendix A). In general, type 1 structures tend to be more stable than type 2 structures, which is expected due to their respective spatial arrangements. Finally, β diesters are more stable in syn conformation, while α diesters prefer the anti conformation.

In step 2, the most stable diesters of each combination of conformation and type (see Table 1) were used to build their respective G2. The structure of a G2 can vary with each pair of guanosines that bind through hydrogen bonds (Figure 5): guanosine **E** can bind to guanosine **F** or **C**; or guanosine **F** can bind to guanosine **D**, forming three different structures of G2 for each diester (here referred to as G2_1_, G2_2_, and G2_3_). Every G2 was constructed by adding two diesters with the same anomers, conformation, and type, e.g., two Aα_2_, two Sγ_2_, etc. Given the larger size of the structures, a smaller basis set (3-21G) was used for optimization. Single point energies were calculated with 6-31G** to improve accuracy. The relative stabilities of anti and syn G2 structures are summarized in Table 2, where **Aα_2_-G2_2_** and **Sγ_2_-G2_3_** are the most stable G2 (Figure 4B; for individual geometry-optimized structures of every G2 see Appendix A).

Optimization shows that G2 structures built from α and γ diesters tend to bend in on themselves, creating ‘*crown-like*’ structures like the ones shown in Figure 4B (see also Appendix A). Therefore, it seems unlikely this type of G2 could lead to G4-stacking and hydrogel formation. The results suggest that structures with only α-guanosines are more stable with anti conformation, while β-guanosine structures prefer the syn conformation.

G3 optimization was narrowed down to six structures, namely, **Aα-G3**, **Aβ-G3**, **Aγ-G3**, **Sα-G3**, **Sβ-G3**, and **Sγ-G3**, which were built by adding a diester to the most stable G2 (see Table 2), respectively, as follows:(1)**Aα_2_-G2_2_** + **Aα_2_** −> **Aα-G3;**(2)**Aβ_1_-G2_1_** + **Aβ_1_** −> **Aβ-G3**;(3)**Aγ_1_-G2_1_** + **Aγ_1_** −> **Aγ-G3;**(4)**Sα_1_-G2_3_** + **Sα_1_** −> **Sα-G3;**(5)**Sβ_1_-G2_1_** + **Sβ_1_** −> **Sβ-G3;**(6)**Sγ_2_-G2_3_** + **Sγ_2_** −> **Sγ-G3.**

In this case, the presented relative energies were calculated relative to **Sβ-G3** (Table 3).

The second most stable G3 obtained was **Aα-G3**, by a ΔE of 2.89 kcal/mol. While this G3 bent inward, forming a *‘two-layered’* type of structure (Appendix A), the opposite was observed for **Sβ-G3**, which spread out across a plane (Figure 4C; for individual geometry-optimized structures of each G3, see Appendix A). Though, overall, anti G3 structures tended to be more stable than their analogue syn counterparts.

Lastly, a fourth diester was added to each G3 to build a G4 (Figure 2). In this case, all G3 structures listed in Table 3 were utilized:(1)**Aα-G3** + **Aα_2_** −> **Aα-G4;**(2)**Aβ-G3** + **Aβ_1_** −>; **Aβ-G4;**(3)**Aγ-G3** + **Aγ_1_** −> **Aγ-G4;**(4)**Sα-G3** + **Sα_1_** −> **Sα-G4;**(5)**Sβ-G3** + **Sβ_1_** −> **Sβ-G4;**(6)**Sγ-G3** + **Sγ_2_** −> **Sγ-G4.**

Table 4 summarizes the relative energies of optimized G4, relative to the most stable structure, **Sβ-G4** (Figure 4D; for geometry-optimized structures of every G4 see Appendix A).

While **Aγ-G4** (containing a mixture of anti α- and β-guanosines) is the second most stable structure (Table 4), it tends to bend and form a *‘bowl-like’* G4-quartet (Figure 6A). The same occurred with **Sγ-G4** (with a mixture of syn α- and β-anomers; Appendix A), which is the least stable of the G4 structures (Table 4). Moreover, **Aα-G4** was the only structure that did not form a closed quartet (Appendix A). On the other hand, **Sβ-G4** (the most stable G4) formed a structure containing a central planar geometry with a hollow (compare Figure 6B with Figure 1), which could possibly promote π-π stacking between quartets.

While the energy difference between α and β anomers of nucleosides tends to be 3–4 kcal/mol [47,48] there are not many experimental studies that focus on the thermodynamic stability of guanosine–borate esters/diesters or G4-quartet anomers (most studies focus on G4-quartet stacking sequences [49]). The stability of G4-quartets also depends on many factors, like temperature, concentration, central cation, the presence of borate, etc. [2,3], so comparisons could be misleading. However, as mentioned previously, it has never been observed that α-nucleosides form G4-quartets or hydrogels [36], and the method used herein gives similar results to what has been experimentally observed.

As previously mentioned, the four carbonyl oxygen atoms in the central hollow of a G4-quartet tend to coordinate with an alkali cation [3]. Mulliken and Löwdin atomic charge distributions in the central hollow of **Sβ-G4** (Figure 7A) show that the nitrogen and oxygen atoms have a large negative charge, behaving as electron donors. The remaining carbon and hydrogen atoms exhibit positive charges, and hence are acceptors. The net negative charge suggests that an alkali cation could coordinate at the site. Since it has been previously shown that K^+^ is the most efficient alkali cation to enhance the stability and robustness of G4-quartets [3], K^+^ was added to the center of **Sβ-G4** (Figure 7B).

Herein, K^+^ coordinated with the four carbonyl ligands at the center of **Sβ-G4**, in a position behind the quartet plane (see Appendix A). When binding to organic molecules, K^+^ tends to have a coordination number of 8 [50]. When coordinated with a **Sβ-G4** and four molecules of water, the optimized structure presented a square antiprismatic-like geometry (Figure 8). This suggests K^+^ could coordinate with the other four oxygen atoms of another quartet. This is typical of guanosine hydrogels, where K^+^ is usually sandwiched between G4-quartets, stabilizing them and creating column-like aggregates [51]. This could not happen in the case of the *‘bowl-like’* γ-G4 structures (Figure 6A). Another interesting detail is that the remaining guanosines of the four diesters are spread out in space and as distant as possible from the center of the G4. In case of self-assembly, these could more easily bind to lateral borate esters, forming entangled fibers that could possibly create a hydrogel.

## 3. Materials and Methods

All initial structures (diesters, G2, G3, and G4) were constructed with Chemcraft version 1.8 [52]. All theoretical calculations were of the DFT type, carried out with GAMESS-US version R3 [53], using the implemented version of the B3LYP functional [54]. A 6-31G** basis set was used for optimization of smaller structures (diesters), and calculation of single point energies of larger structures (G2, G3, and G4) after optimization with a 3-21G basis set. Mulliken and Löwdin atomic charges were performed with Chemcraft version 1.8 [52].

## 4. Conclusions

As previously stated, guanosine is prevalently found in biological systems as the anomer β-guanosine. While this anomer tends to self-assemble into G4-quartets and subsequently hydrogels, the same does not seem to happen with the α anomer [17,36]. This study proposes that the self-assembling properties of borate–guanosine hydrogels could have led to the prebiological separation of the two anomers and subsequent selective synthesis of guanosine in living beings.

The results suggest that borate-β-guanosine diesters, in syn conformation, form the most stable G4-quartets. The 3D organization and stereochemical environment formed by this anomer promotes coordination with K^+^ and π-π stacking between the quartets, possibly favoring the formation of entangled fibers and enabling the self-assembly of a hydrogel. As previously thought, this does not seem to happen if the α-anomer is present: **Aα-G4** does not form a closed quartet and both γ-G4s tend to have *‘bowl-like’* structures. Given the properties of borate–guanosine hydrogels (thermal robustness over a large temperature range and tendency to partially self-assemble, even when guanosine is phosphorylated) [17], it is possible these structures could have played a role in the prebiological separation of the two anomers. Their external layers could have prevented the degradation of β-guanosine, protecting it, stabilizing it, concentrating it over time, and ultimately, giving it an advantage over α-guanosine.

These results suggest that borate–guanosine hydrogels would have selected β-guanosine as a component of current biological structures. As previously suggested [17], the formation of hydrogel globules after guanosine phosphorylation would have led to the formation of proto-cellular systems that would have facilitated the confinement of prebiological molecules, acting as cradles for prebiotic evolution. As mentioned previously, these hydrogels can confine and protect other ribonucleosides and serve as media for the synthesis of ribonucleotides [17]. Experimental data show that supramolecular assemblies, like the ones formed by melamine and barbituric acid, tend to preferentially incorporate and increase the number of β-nucleotides over α-nucleotides [55]. A similar phenomenon could have happened in borate–guanosine hydrogels, leading to an enrichment of the β-anomer of other canonical ribonucleosides/ribonucleotides.

On the other hand, it would also be interesting to apply the method herein described to other ribonucleosides, since, at least, β-adenosine and β-cytidine can form hydrogels [56,57]. Could these self-assembling properties have led to the separation of other nucleoside anomers? Interestingly, D-sugars also predominate in nature (such as D-ribose, the sugar frame in RNA). Perhaps elucidating the structures of hydrogels composed of borate and D- and L-ribonucleosides would also help to understand this phenomenon, since they have different reactivity [46].

## Figures and Tables

**Figure 1 ijms-26-12103-f001:**
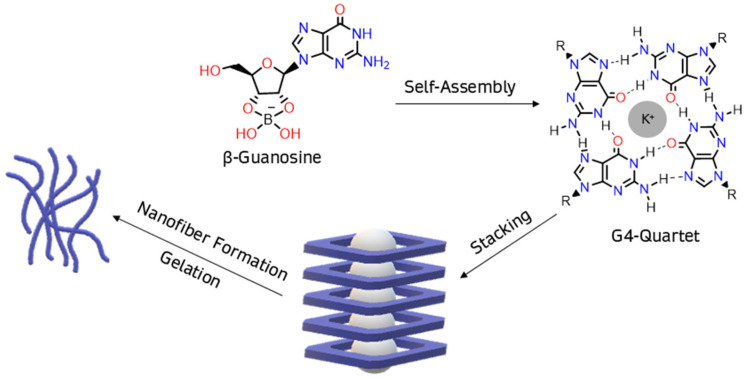
In the presence of borate, β-guanosine self-assembles into G4-quartets, forming a hydrogel.

**Figure 2 ijms-26-12103-f002:**
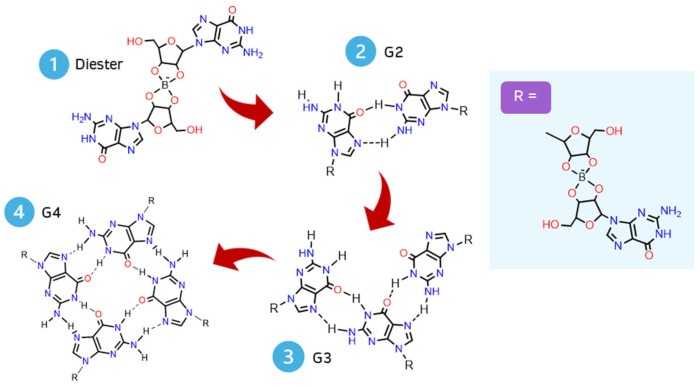
Scheme of a step-by-step geometry optimization of a G4-quartet.

**Figure 3 ijms-26-12103-f003:**
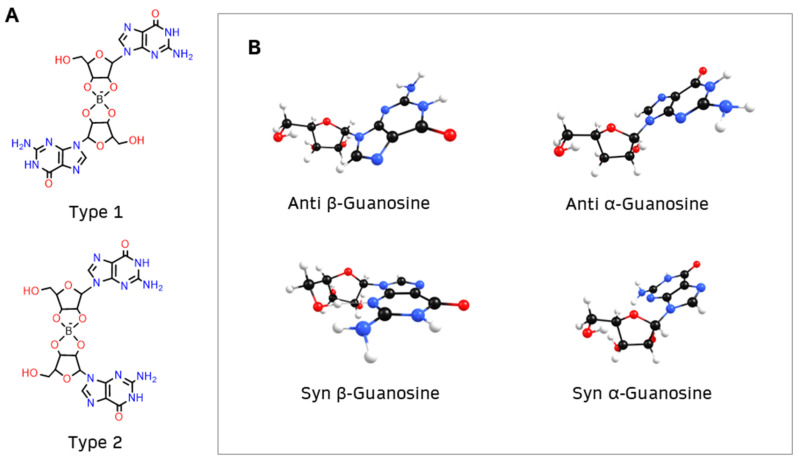
(**A**) Borate binds to guanosine, forming diesters with two possible structures: type 1, where the nucleobases of guanosine are opposite to each other, or type 2, where the nucleobases are on the same side of the diester; (**B**) β-guanosine (**left**) and α-guanosine (**right**) can have anti or syn conformation.

**Figure 4 ijms-26-12103-f004:**
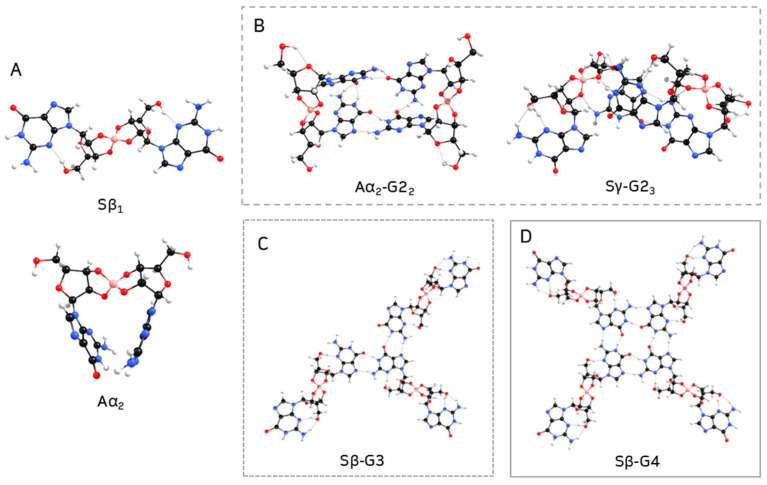
Geometry-optimized structures of most stable borate–guanosine (**A**) diesters; (**B**) G2; (**C**) G3; and (**D**) G4.

**Figure 5 ijms-26-12103-f005:**
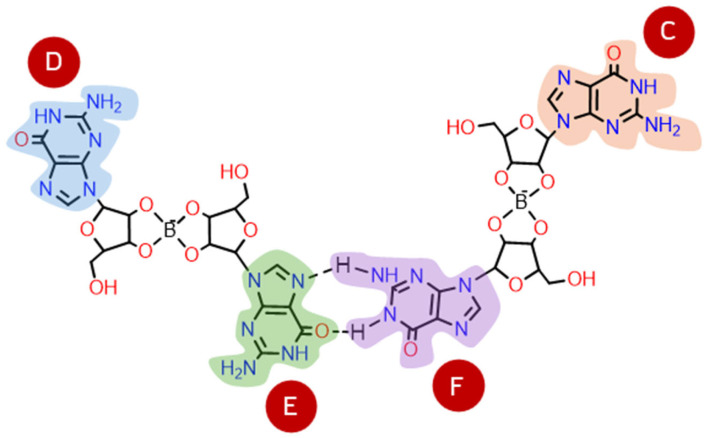
G2 structures are formed by two borate–guanosine diesters, where guanosine **E** can bind to guanosine **F** or **C**; or guanosine **F** can bind to guanosine **D**, forming three possible structures of G2 for each type of diester (G2_1_, G2_2_, and G2_3_).

**Figure 6 ijms-26-12103-f006:**
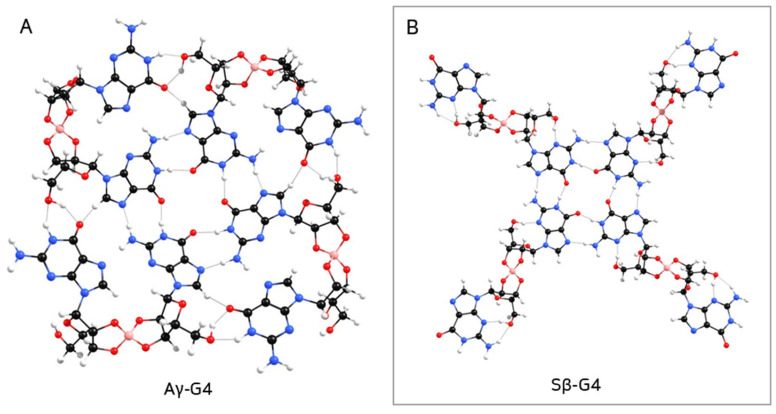
Geometry-optimized structure of (**A**) **Aγ-G4**, forming a *‘bowl-like’* G4-quartet, which would prevent the formation of column-like aggregates; (**B**) **Sβ-G4**.

**Figure 7 ijms-26-12103-f007:**
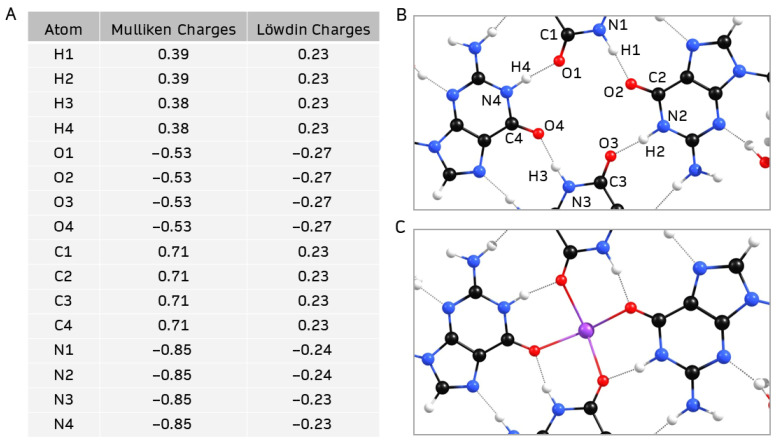
(**A**,**B**) Mulliken and Löwdin charge distribution of the central hollow of geometry-optimized **Sβ-G4**; (**C**) central hollow of geometry-optimized structure of **Sβ-G4** coordinated with K^+^.

**Figure 8 ijms-26-12103-f008:**
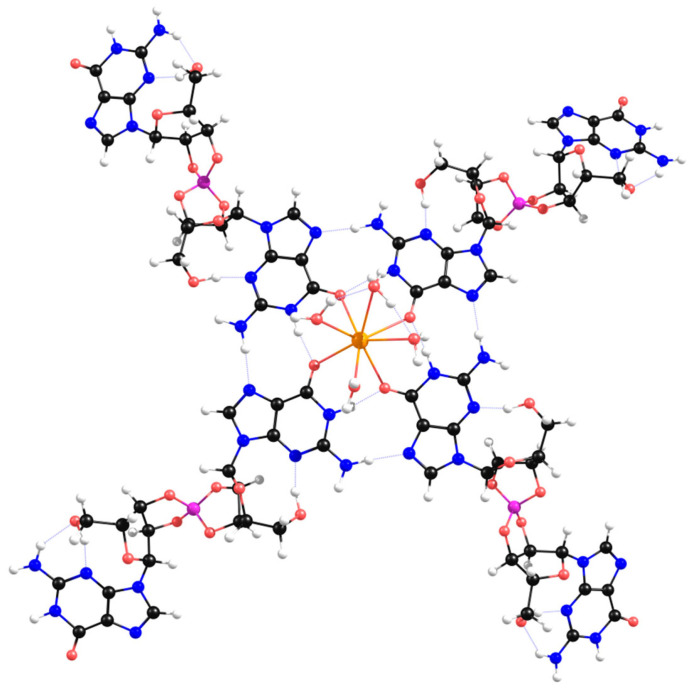
Geometry-optimized structure of **Sβ-G4** coordinated with K^+^ and four molecules of water.

**Table 1 ijms-26-12103-t001:** Relative energies ΔE (in kcal/mol) of α, β, and γ diesters of types 1 and 2, with anti (A) and syn (S) conformation (where **Aα_2_** and **Sβ_1_** are the most stable diesters, respectively). The listed values result from DFT (B3LYP/6-31G**) calculations.

Anti Conformation		Syn Conformation	
Diester	ΔE (kcal/mol)	Diester	ΔE (kcal/mol)
Aα_1_	7.74	Sα_1_ *	12.90
Aα_2_ *	0.00	Sα_2_	21.13
Aβ_1_ *	9.36	Sβ_1_ *	0.00 *
Aβ_2_	10.60	Sβ_2_	1.20
Aγ_1_ *	2.11	Sγ_1_	6.53
Aγ_2_	9.71	Sγ_2_ *	2.91

* Most stable diesters of each combination of anomers, conformation, and type; used to build respective G2 structures.

**Table 2 ijms-26-12103-t002:** Relative energies ΔE (in kcal/mol) of α-, β-, and γ-G2 structures, with anti and syn conformation (where **Aα_2_-G2_2_** and **Sγ_2_-G2_3_** are the most stable structures, respectively). The listed values result from DFT (B3LYP) optimization with 3-21G basis set and single point energy calculation with 6-31G** basis set.

Anti Conformation ΔE (kcal/mol)			Syn Conformation ΔE (kcal/mol)		
G2 *	OPT (3-21G)	SPE (6-31G**)	G2 *	OPT (3-21G)	SPE (6-31G**)
Aα_2_-G2_1_	9.16	7.03	Sα_1_-G2_1_	27.58	19.28
Aα_2_-G2_2_ **	0.00	0.00	Sα_1_-G2_2_	27.55	19.16
Aα_2_-G2_3_	0.08	0.19	Sα_1_-G2_3_ **	27.47	19.12
Aβ_1_-G2_1_ **	36.36	32.06	Sβ_1_-G2_1_ **	6.57	5.63
Aβ_1_-G2_2_	41.85	32.26	Sβ_1_-G2_2_	7.32	5.77
Aβ_1_-G2_3_	40.08	32.19	Sβ_1_-G2_3_	8.33	6.43
Aγ_1_-G2_1_ **	9.40	11.07	Sγ_2_-G2_1_	0.63	6.22
Aγ_1_-G2_2_	19.24	14.02	Sγ_2_-G2_2_	14.79	7.16
Aγ_1_-G2_3_	26.05	19.15	Sγ_2_-G2_3_ **	0.00	0.00

* γ-G2_1_, γ-G2_2_, and γ-G2_3_ structures have hydrogen bonds between β…β, α…β, and α…α guanosines, respectively; ** Most stable G2 of each combination of anomers, conformation, and type; used to build respective G3 structures.

**Table 3 ijms-26-12103-t003:** Relative energies ΔE (in kcal/mol) of α-, β-, and γ-G3 structures, with anti and syn conformation (**Sβ-G3** is the most stable structure). The listed values result from DFT (B3LYP) optimization with 3-21G basis set and single point energy calculation with 6-31G** basis set.

Anti Conformation ΔE (kcal/mol)			Syn Conformation ΔE (kcal/mol)		
G3	OPT (3-21G)	SPE (6-31G**)	G3	OPT (3-21G)	SPE (6-31G**)
Aα-G3	2.89	9.40	Sα-G3	38.56	29.54
Aβ-G3	42.76	10.91	Sβ-G3	0.00	0.00
Aγ-G3	9.77	10.91	Sγ-G3	27.62	32.23

**Table 4 ijms-26-12103-t004:** Relative energies ΔE (in kcal/mol) of α-, β-, and γ-G4 structures, with anti and syn conformation (where **Sβ-G4** was the most stable G4), resulting from DFT (B3LYP) optimization with 3-21G basis set and single point energy calculation with 6-31G** basis set.

Anti Conformation ΔE (kcal/mol)			Syn Conformation ΔE (kcal/mol)		
G4	OPT (3-21G)	SPE (6-31G**)	G4	OPT (3-21G)	SPE (6-31G**)
Aα-G4	36.26	3.42	Sα-G4	64.20	59.11
Aβ-G4	54.15	15.68	Sβ-G4	0.00	0.00
Aγ-G4	11.37	0.22	Sγ-4	73.44	78.01

## Data Availability

The original contributions presented in this study are included in the article/Appendix A. Further inquiries can be directed to the corresponding authors.

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
