# Peer review of "Borate–Guanosine Hydrogels and Their Hypothetical Participation in the Prebiological Selection of Ribonucleoside Anomers: A Computational (DFT) Study"

_ijms, 2025, doi:10.3390/ijms262412103_

Round 1
Reviewer 1 Report
Comments and Suggestions for Authors
Manuscript ID: ijms-3989856
Title: Borate-Guanosine Hydrogels and Their Hypothetical Participation in the Prebiological Selection of Ribonucleoside Anomers: A Computational (DFT) Study.
Franco A. et al. simulated the ribonucleoside guanosine a- and b-anomers in borate-guanosine hydrogels by computational DFT method. The authors concluded that only b-anomer formed most stable planar G4-quartet with borate, capable of self-assembling into a hydrogel. The structure of containing up to 4 borate bound guanosine diesters of two types having a- and b-guanosine anomers in anti- and syn-conformation were optimized using 6-31G** and 3-21G basis sets.
Unfortunately some discussions are missing in the manuscript:
1) It was not clearly stated whether the selected computational technique is able to distinguish relative small energy difference between a- and b-anomers (for sugars often on the order of 1 kcal/mol) in the final G4-quartet. For the systems having multiple anomers (8 in guanosine tetramer) the precision might be about ~0.1 kcal/mol for the least precise technique employed (3-21G basis sets).
2) As the relative stability of anomers depends on the solvent (while the a-anomer has usually lower energy in vacuum, the b-anomer in water), the conclusions made seems to be limited only to the vacuum condition. Whether and how they can be extended to aqueous environment of G4-quartet is missing.
3) The protecting role of entitled hydrogels ‘prevented degradation of compounds by aggressive primitive radiation’ was concluded from the results estimated in water–free condition, which is different compare to published (J. Phys. Chem. Lett. 2018, 9, 4981, doi: 10.1021/acs.jpclett.8b02077) spontaneous formation of ribonucleotides reaction modelled under realistic prebiotic conditions.
This paper is recommended for the publication after major revision taking these aspects into account.
Author Response
For research article
|
Response to Reviewer 1 Comments
|
||
|
1. Summary |
|
|
|
Thank you very much for taking the time to review this manuscript. Please find detailed responses below and the corresponding revisions/corrections highlighted/in track changes in the re-submitted files.
|
||
|
2. Questions for General Evaluation |
Reviewer’s Evaluation |
Response and Revisions |
|
Does the introduction provide sufficient background and include all relevant references? |
Can be improved |
Please see the point-by-point comment section. |
|
Is the research design appropriate? |
Must be improved |
Please see the point-by-point comment section. |
|
Are the methods adequately described? |
Can be improved |
Please see the point-by-point comment section. Please see the point-by-point comment section. |
|
Are the results clearly presented? |
Can be improved |
Please see the point-by-point comment section. |
|
Are the conclusions supported by the results? |
Can be improved |
Please see the point-by-point comment section. |
|
Are all figures and tables clear and well-presented? |
Can be improved |
We believe this may be a problem caused by the required template. We can send the main text in another format, if needed. |
|
3. Point-by-point response to Comments and Suggestions for Authors |
||
|
Comment 1: It was not clearly stated whether the selected computational technique is able to distinguish relative small energy difference between a- and b-anomers (for sugars often on the order of 1 kcal/mol) in the final G4-quartet. For the systems having multiple anomers (8 in guanosine tetramer) the precision might be about ~0.1 kcal/mol for the least precise technique employed (3-21G basis sets).
|
||
|
Response 1: Thank you for pointing this out. Although optimizations were carried out at 3-21G level (which usually predicts accurate structures) the energies were obtained in single point calculations at 6-31G** to improve accuracy. Ideally, we would like to compare the calculated results with experimental data. While the energy difference between α and β anomers of nucleosides is approximately 3 - 4 kcal/mol (Helyion 2022, 6, e09657, doi: 10.1016/j.heliyon.2022.e09657; Theor. Chem. Acc. 2008, 120, 215-241, doi: 10.1007/s00214-007-0310-x) there is not enough experimental data regarding nucleoside-borate esters/diesters or G4-quartet anomers (of which, most thermodynamical studies focus on their many possible stacking sequences). (ChemBioChem 2021, 22, 2848-2856, doi: 10.1002/cbic.202100127)). The stability of G4-quartets also depends on many factors, like temperature, concentration, central cation, the presence of borate, etc., so comparisons could be misleading. Additionally, it has never been observed experimentally that α-nucleosides form G4-quartets or hydrogels (RSC Adv. 2019, 9, 14302-14320, doi: 10.1039/C9RA01399G). Our manuscript tries to explain this phenomenon based on the G4 structure of each anomer.
We added the following sentences to the manuscript [Page 8, Lines 214-221, marked in red] and references 36, 47, 48 and 49 to the list [Page 13, Lines 420-426, marked in red] to the list: While the energy difference between the α and β anomers of nucleosides tend to ca. 3 - 4 kcal/mol [47,48] there are not many experimental studies that focus on the thermodynamic stability of guanosine-borate esters/diesters or G4-quartet anomers (most studies focus on G4-quartet stacking sequences [49]). The stability of G4-quartets also depends on many factors, like temperature, concentration, central cation, the presence of borate, etc., [2,3] so comparisons could be misleading. However, as afore-mentioned, it has never been observed that α-nucleosides form G4-quartets or hydrogels [36], and the method herein used gives similar results to what is experimentally observed. |
||
|
Comments 2: As the relative stability of anomers depends on the solvent (while the a-anomer has usually lower energy in vacuum, the b-anomer in water), the conclusions made seem to be limited only to the vacuum condition. Whether and how they can be extended to aqueous environment of G4-quartet is missing.
|
||
|
Response 2: Castaneda and Matta (Helyion 2022, 6, e09657, doi: 10.1016/j.heliyon.2022.e09657) observed that the β anomer of guanosine is slightly more stable than the α anomer in vacuum phase. Regardless, we agree with this comment. However, the computational time to add PCM (Polarizable Continuum Model) with SMD (Solvation Model Density) to add corrections for cavitation, dispersion and solvent structure in a model as large as the G4 makes the study in aqueous environment not achievable within our current computational capabilities. As so we decided to restrict the calculations to vacuum and be cautious in our extrapolation to aqueous environment.
|
||
Comment 3: The protecting role of entitled hydrogels ‘prevented degradation of compounds by aggressive primitive radiation’ was concluded from the results estimated in water–free condition, which is different compare to published (J. Phys. Chem. Lett. 2018, 9, 4981, doi: 10.1021/acs.jpclett.8b02077) spontaneous formation of ribonucleotides reaction modelled under realistic prebiotic conditions.
Response 3: Thank you for pointing this out. While calculations were carried out without a PCM, explicit waters were included in the calculations to show the viability of the hydrogels. The suggestion that hydrogels may prevent degradation by radiation is based on experimental data. Hydrogels act as a physical barrier due to their ability to encapsulate water and many studies show they may protect against radiation (e.g., Gels 2023, 9, 301, doi:10.3390/gels9040301; Biomimetics 2025, 10, 758; doi: 10.3390/biomimetics10110758; Polymers 2025, 17, 2234, doi:10.3390/polym17162234; Sci. Rep. 2020, 10, 21689, doi:10.1038/s41598-020-78663-x), including derivates of guanosine (J. Mater. Chem. B 2025, 13, 3039-3048, doi: 10.1039/D4TB02380C). It also has been suggested that prebiological hydrogels may have acted in a similar manner, since they can self-assemble in the natural world, on rock and mineral surfaces (J. Colloid Interface Sci. 2014, 431, 250-254, doi:10.1016/j.jcis.2014.02.034; ChemSystemsChem 2025, e00038, doi:10.1002/syst.202500038). We included these references in the main text.
References Gels 2023, 9, 301, doi:10.3390/gels9040301; Biomimetics 2025, 10, 758; doi: 10.3390/biomimetics10110758; Polymers 2025, 17, 2234, doi:10.3390/polym17162234; Sci. Rep. 2020, 10, 21689, doi:10.1038/s41598-020-78663-x; J. Colloid Interface Sci. 2014, 431, 250-254, doi:10.1016/j.jcis.2014.02.034; ChemSystemsChem 2025, e00038, doi:10.1002/syst.202500038 were added to Page 2, Line 43 of the manuscript, as examples of biological applications of hydrogels and added to the Reference List as references number [25-31], respectively (marked in the manuscript in red.
Reviewer 2 Report
Comments and Suggestions for Authors
In this study, the authors performed DFT calculations on α and β anomers of guanosine to investigate the origin of anomeric selectivity relevant to the RNA world hypothesis. Their computations traced the structural optimization from borate-guanosine diesters up to G4-quartets. The results identified β-syn-guanosine as the most stable conformation and, critically, as the only anomer capable of forming a planar G4-quartet with borate. This specific quartet is essential for the self-assembly of borate-guanosine hydrogels. The authors therefore propose that these stable hydrogels, which could have acted as protective prebiotic frameworks by shielding early molecules from radiation and degradation, conferred a selective advantage.
This work can be interesting for both computational and experimental chemistry community. I would like to ask the authors to address the comments below.
- My main concern is the DFT functional used in this work. B3LYP is a good choice. However, the dispersion correction was not added. Since the studied systems have conjugated structures, the the long-range dispersion interaction can be important in the studied system. Can the authors estimate the error of not adding dispersion correction?
- Another concern is that the 3-21G basis set is too small to give reasonable results. To get reasonable optimized geometries, at least 6-31G* or cc-pVDZ level basis sets are needed.
- The solvation effect can be important that re-arranges the electron density from solvated molecules to solvent molecules. Did the authors use any implicit solvation model to cover the solvation effect? In addition, more explicit water molecules might be needed to give converged solvation effect.
- It would be helpful to also calculate the thermodynamics such as free energy change in the conformation change process.
- Is there a way to characterize the non-covalent interactions in the studied systems? For example, using the Non-covalent interaction (NCI) plots software to visualize it?
- The reference for the B3LYP functional is missing: Phys. Rev. B 37, 785.
Author Response
For research article
|
Response to Reviewer X Comments
|
||
|
1. Summary |
|
|
|
Thank you very much for taking the time to review this manuscript. Please find the detailed responses below and the corresponding revisions/corrections highlighted/in track changes in the re-submitted files.
|
||
|
2. Questions for General Evaluation |
Reviewer’s Evaluation |
Response and Revisions |
|
Does the introduction provide sufficient background and include all relevant references? |
Can be improved |
Thank you for your evaluation. In your opinion, how can we improve the introduction? |
|
Is the research design appropriate? |
Must be improved |
Please see the point-by-point comment section. |
|
Are the methods adequately described? |
Yes |
|
|
Are the results clearly presented? |
Yes |
|
|
Are the conclusions supported by the results? |
Must be improved |
Please see the point-by-point comment section. |
|
Are all figures and tables clear and well-presented? |
Yes |
|
|
3. Point-by-point response to Comments and Suggestions for Authors |
||
|
Comment 1: My main concern is the DFT functional used in this work. B3LYP is a good choice. However, the dispersion correction was not added. Since the studied systems have conjugated structures, the long-range dispersion interaction can be important in the studied system. Can the authors estimate the error of not adding dispersion correction?
|
||
|
Response 1: Thank you for pointing this out. Unfortunately, due to the large number of atoms and orbitals involved in the G4-quartets, it is not feasible to add wan der Waals D3-type dispersion corrections, in a reasonable time, with our current computational capabilities.
|
||
|
Comment 2: Another concern is that the 3-21G basis set is too small to give reasonable results. To get reasonable optimized geometries, at least 6-31G* or cc-pVDZ level basis sets are needed. |
||
|
Response 2: We appreciate this comment. Although optimizations were carried out at 3-21G level (which we think usually predicts accurate structures) the energies were obtained in single point calculations at 6-31G** to improve accuracy. Ideally, we would like to use a larger basis set, but as we mentioned in response 1, this is not feasible, with our current computational capabilities. We would also have liked to compare the calculated results with experimental data. While the energy difference between the α and β anomers of nucleosides is approximately 3 - 4 kcal/mol (Helyion 2022, 6, e09657, doi: 10.1016/j.heliyon.2022.e09657; Theor. Chem. Acc. 2008, 120, 215-241, doi: 10.1007/s00214-007-0310-x) there is not enough experimental data regarding nucleoside-borate esters/diesters or G4-quartets (of which, most thermodynamical studies focus on their many possible stacking sequences, instead of single quartets. (ChemBioChem 2021, 22, 2848-2856, doi: 10.1002/cbic.202100127)). The stability of G4-quartets also depends on many factors, like temperature, concentration, central cation, the presence of borate, etc., so comparisons could be misleading. Additionally, it has never been observed experimentally that α-nucleosides form G4-quartets or hydrogels (RSC Adv. 2019, 9, 14302-14320, doi: 10.1039/C9RA01399G). Our manuscript tries to explain this phenomenon based on the G4 3D-structure of each anomer.
We added the following sentences to the manuscript [Page 8, Lines 214-221, marked in red] and references 36, 47, 48 and 49 to the list [Page 13, Lines 420-426, marked in red] to the list: While the energy difference between α and β anomers of nucleosides tend to be around 3 - 4 kcal/mol [47,48] there are not many experimental studies that focus on the thermodynamic stability of guanosine-borate esters/diesters or G4-quartet anomers (most studies focus on G4-quartet stacking sequences [49]). The stability of G4-quartets also depends on many factors, like temperature, concentration, central cation, the presence of borate, etc., [2,3] so comparisons could be misleading. However, as afore-mentioned, it has never been observed that α-nucleosides form G4-quartets or hydrogels [36], and the method herein used gives similar results to what is experimentally observed.
Comment 3: The solvation effect can be important that re-arranges the electron density from solvated molecules to solvent molecules. Did the authors use any implicit solvation model to cover the solvation effect? In addition, more explicit water molecules might be needed to give converged solvation effect.
Response 3: Thank you for your comment. The explicit water molecules were not included to simulate the solvation effects, but to test the viability of water molecules as ligands in producing a stable center. To expand on our earlier responses, the computational time to add PCM (Polarizable Continuum Model) with the SMD (Solvation Model Density) to add corrections for cavitation, dispersion and solvent structure in a model as large as the G4 makes the study in aqueous environment not achievable within our current computational capabilities. As so we decided to restrict the calculations to vacuum and be cautious in our extrapolation to aqueous environment that are according to the experimental G4 anomers of guanosine.
Comment 4: It would be helpful to also calculate the thermodynamics such as free energy change in the conformation change process.
Response 4: We believe our early responses also apply to this comment. Unfortunately, due to the large number of atoms and orbitals involved in G4-quartets we cannot do these calculations with our current computational capabilities, in a reasonable amount of time.
Comment 5: Is there a way to characterize the non-covalent interactions in the studied systems? For example, using the Non-covalent interaction (NCI) plots software to visualize it?
Response 5: We appreciate this suggestion. We have tried to characterize the non-covalent interactions through the Multiwfn software, but it seems GAMESS-US output files do not print expansion coefficients of all orbitals for the final geometry. Regretfully, we would not be able to obtain the plots without further calculations and, as mentioned above, this type of calculation would be extremely difficult to perform given our present computational capabilities.
Comment 6: The reference for the B3LYP functional is missing: Phys. Rev. B 37, 785.
Response 6: Thank you for reminding us. The missing reference was included in Page 9, Line 254, as reference number [54]. We also added it to the Reference List [Page 13, Lines 440-441], marked in red in the manuscript.
|
||
Round 2
Reviewer 2 Report
Comments and Suggestions for Authors
This manuscript has been improved after revisions. No further review is needed.